

# Sleep duration has a limited impact on the prevalence of menstrual irregularities in athletes: a cross-sectional study

Mana Miyamoto[1,2] and Kenichi Shibuya[1,2]

[1] Department of Health and Nutrition, Niigata University of Health and Welfare, Niigata, Japan
[2] Graduate School of Health and Welfare, Niigata University of Health and Welfare, Niigata, Japan

## ABSTRACT

**Background:** This study aimed to investigate the relationship between the prevalence of menstrual irregularities, energy intake, and sleep deprivation among female athletes.

**Methods:** A total of 128 female athletes, with an average age of $19.2 \pm 1.2$ years, participated in the study and tracked their food intake over a three-day period. Menstrual status and sleep duration were assessed using a questionnaire, and psychological anxiety was evaluated using the State and Trait Anxiety Inventory (STAI). These were measured once during the investigation. The impact of sleep status on state anxiety and daily energy intake was examined using the T-test. A generalized linear model (GLM) with a log link function was employed to investigate the effects of sleep deprivation on the presence of menstrual irregularities.

**Results:** As the results of the present study, sleep deprivation significant increased both state and trait anxiety ($p < 0.05$), as well as affecting energy intake ($p < 0.05$), particularly protein and carbohydrate intakes ($p < 0.05$). However, GLM analysis indicated that while sleep deprivation did not directly influence the prevalence of menstrual irregularities ($p > 0.05$), state anxiety emerged as a significant factor impacting the prevalence of menstrual irregularities ($p < 0.05$).

**Conclusions:** The results of the present study suggest a potential pathway wherein sleep deprivation might elevate state anxiety levels, consequently indirectly contributing to an increase the probability of menstrual irregularities. In conclusion, the results of the presents study provide novels insights suggesting that sleep deprivation might directly increase state anxiety and indirectly affect the prevalence of menstrual irregularities. Hence, decreased sleep duration might be related to mental health issues and the prevalence of menstrual irregularities both significant concerns among female athletes. Future studies will play a crucial role in further elucidating how sleep patterns impact the health and well-being of female athletes.

Corresponding author
Mana Miyamoto,
mana-miyamoto@nuhw.ac.jp

# INTRODUCTION

Sleep constitutes an indispensable aspect of daily life, intricately linked with overall health (*Sabo et al., 1991*; *Tsuno, Besset & Ritchie, 2005*). Its role in fostering cell growth and enhancing immune function underscores its critical importance for overall well-being.
In addition, melatonin, the primary hormone secreted during sleep, regulates biorhythms (*Turek & Gillette, 2004*). Consequently, sleep might significantly influence sports performance (*Watson, 2017*).

Conversely, menstrual irregularities, a component of the female athlete triad, hold global significance (*Nattiv et al., 2007*; *Mountjoy et al., 2014*). Factors contributing to menstrual irregularities among female athletes commonly include insufficient energy intake, rapid weight loss, and reduced body fat (*Enea et al., 2011*). Moreover, previous research has shown the role of insufficient nutritional intake, specifically deficiencies in carbohydrates and vitamin D, in the prevalence of menstrual irregularities among female athletes (*Miyamoto, Hanatani & Shibuya, 2021a*, *2022*).

Additionally, mounting evidence suggests an association between the prevalence of menstrual irregularities and psychological distress, characterized by elevated levels of depression and psychological anxiety (*Schliep et al., 2015*; *Breen et al., 2005*; *Kim et al., 2018*; *Miyamoto, Hanatani & Shibuya, 2021a*, *2021b*, *2022*; *Yamamoto et al., 2009*). A recent study reported that increased psychological anxiety may increase the risk of menstrual irregularities in elite female athletes (*Miyamoto, Hanatani & Shibuya, 2021b*). Indeed, top-level athletes are known to exhibit higher levels of psychological anxiety compared to the general population (*Miyamoto, Hanatani & Shibuya, 2021b*; *Gulliver et al., 2015*). A study involving 224 elite Australian athletes found that 46.4% of them were experiencing symptoms of at least one of the mental health problems (*Gulliver et al., 2015*). Notably, elevated psychological anxiety is associated with an increased risk of menstrual irregularity in elite female athletes (*Miyamoto, Hanatani & Shibuya, 2021a*, *2021b*). Several prior studies have also suggested that psychological stress can affect menstrual function and contribute to menstrual irregularities (*Yamamoto et al., 2009*; *Kloss et al., 2015*). These reports indicate that psychological anxiety may play a significant role in the cause of menstrual irregularities in athletes. Therefore, reducing psychological anxiety in female athletes is important for maintaining healthy reproductive function.

Furthermore, recent evidence suggests that sleep disturbances, including shortened sleep duration and insomnia, may contribute to menstrual cycle alterations and the development of menstrual disorders and also suggested that individuals who sleep 5 h or less have a higher risk of experiencing irregular menstruation compared to those who sleep 6 h or more (*Kim et al., 2018*). Additionally, another study indicated that short sleep duration was associated with changes in menstrual cycle (*Lim et al., 2016*). While the impact of mental health problems and sleep deprivation on menstrual function has been acknowledged, it remains unclear which factor exerts a stronger influence. Furthermore, there is a shortage of studies investigating the relationship between psychological anxiety, sleep deprivation, and menstrual irregularities among elite/international-level athletes. Psychological anxiety, sleep deprivation, and energy intake may play a combined role in menstrual irregularities. Therefore, this study aims to assess the relative impact of mental anxiety, sleep deprivation, and energy intake on menstrual function in top-level female athletes.

Additionally, previous studies (*Tu et al., 2012*; *Ohida et al., 2001*; *Schmid et al., 2009*) have provided inconclusive findings concerning the relationship between short sleep

duration and energy intake. While some studies suggest a connection, others have found no significant association. In summary, this study aimed to clarify the relative impacts of psychological anxiety, sleep deprivation, and energy intake on menstrual irregularities in elite-level female athletes.

We hypothesize that sleep deprivation might not directly influence the prevalence of menstrual irregularities but could contribute to psychological anxiety and inadequate energy intake, resulting in menstrual irregularities in elite-level female athletes, and that heightened psychological anxiety will be associated with an elevated risk of menstrual irregularities. The intent of the present study was to provide insights that may assist female athletes in the sports settings to maintain healthy reproductive function while continuing to compete.

## METHODS

### Participants

This study utilized a cross-sectional design conducted over 6 months and involved 140 female athletes, aged between 15 and 23 years (19.2 ± 1.2 years). These athletes competed at the inter-college level in track and field events and at the international level in rowing. The participants comprised athletes from the Japanese national rowing team ($n = 50$) and long-distance runners from a university track and field club ($n = 90$), all of whom were at least 1-year post-menarche. All included athletes were naturally menstruating and were not using non-hormonal contraceptives. They willingly participated in the surveys as members of these teams.

The survey was carried out outside the primary competitive period throughout the year. Prior to their involvement, informed consent was obtained from all athletes. However, a total of 12 athletes (eight from track and field and four from rowing) were excluded from the analysis due to incomplete data. The appropriate sample size of 107 was calculated using the following parameters: odds ratio = 2, H0 = 0.15, $\alpha = 0.05$, power = 0.8, and a X distribution for logistic regression.

The study protocol received ethical approval from the Ethics Committee of Niigata University of Health and Welfare (Approval #18147-19032), and all participants provided written informed consent after receiving a comprehensive explanation regarding the study's nature, procedures, and non-invasive approach.

### Dietary intake, body mass, and body composition

Participants employed a meal-recording method to track their dietary intake. Over the course of 3 days within a week, they captured photographs of each meal (breakfast, lunch, dinner, and snacks). These images were accompanied by detailed notes specifying the type and quantity of food consumed, which were then submitted to the team dietitian for analysis. The nutrient content derived from these photos was analyzed by dietitians using the Japan National Nutrient Database.

From this analysis, daily nutrient intake data were computed, encompassing total energy intake (kcal), macronutrients (grads and as a percentage), and fiber intake (grams), averaged on a daily basis. Participants' body mass (BM) and body fat percentage (%BF)

were measured using a commercially available home scale (BC-314; Tanita. Co., Tokyo, Japan) immediately upon waking up.

The %BF data were self-reported by the respondents voluntarily. Body mass index (BMI) was calculated as weight (in kilograms) divided by height squared (in meters). Height, BM, and %BF were measured once after the nutrition investigation, followings procedures used in prior studies (*Miyamoto, Hanatani & Shibuya, 2022*, *2021b*).

## Anxiety, menstrual cycle, and sleep duration

State and trait anxiety levels were evaluated using the Japanese version of the State and Trait Anxiety Inventory (STAI-J), a widely accepted tool for anxiety (*Spielberger, Gorsuch & Lushene, 1970*; *Shimizu & Imae, 1981*). The STAI is recognized for its clinical validity, and a clinical cutoff value of 40 has been established (*Grant, McMahon & Austin, 2008*). Participants' state and trait anxiety were assessed once during the investigation, consistent with the timing utilized in previous studies (*Miyamoto, Hanatani & Shibuya, 2021a*, *2021b*, *2022*). State anxiety referred to a transient response to unfavorable events, while trait anxiety reflected a predisposition to experience concerns and worries across various situations.

Participants completed a questionnaire detailing the calendar days of their menstrual cycles, which included the date of the most recent menstrual period and to record their average weekly sleep duration. This information, along with their dietary intake data, was provided to the dietitians.

For the present study, menstrual irregularities were defined as having an interval of 2 months or more between menstrual cycles, experiencing irregular cycles or amenorrhea (absence of menstruation) for three months or longer.

## Statistical analyses

The data were presented as mean ± standard deviation (S.D.) for descriptive purposes, and statistical results were expressed as mean ± standard error (S.E.) to indicate the precision of estimates. Participants were divided into two groups according to their sleep duration: individuals sleeping less than 6 h per day were categorized into the Short sleeper group (defined as experiencing sleep deprivation (*Kim et al., 2018*; *Krishnan, Noakes & Lyons, 2016*)) while those sleeping more than 6 h per day comprised the Long sleeper group.

The impact of sleep status on state anxiety and daily energy intake was examined using the T-test. A generalized linear model (GLM) with a log link function was employed to investigate the effects of sleep deprivation on the presence of menstrual irregularities. The sleep deprivation data comprised of binary values (0 and 1) representing "normal" and "sleep deprivation". The menstrual irregularity data comprised of binary values (0 and 1) representing "normal menstruation" and "menstrual irregularity" states. GLMs were also used to assess the effects of state anxiety, sleep status, and energy intake (total daily caloric intake) on the prevalence of menstrual irregularities. The selection of independent variables in this analysis, including state anxiety, sleep status, energy intake, and carbohydrate intake, was guided by previous research (*Schliep et al., 2015*; *Kim et al., 2018*),

**Table 1 Mean values of physical characteristics, state, and trait anxiety (Mean ± SD).**

|  | All participants | Long sleeper | Short sleeper | $p$ |
|---|---|---|---|---|
| Height (cm) | 162.6 ± 6.4 | 162.7 ± 6.7 | 161.9 ± 4.4 | 0.451 |
| Body mass (kg) | 54.8 ± 7.2 | 54.9 ± 7.4 | 54.0 ± 6.0 | 0.525 |
| %BF (%) | 19.0 ± 3.4 | 19.0 ± 3.5 | 19.3 ± 2.8 | 0.691 |
| BMI | 20.7 ± 1.8 | 20.7 ± 2.0 | 20.6 ± 1.9 | 0.862 |
| State anxiety | 43.2 ± 0.5 | 42.4 ± 9.6 | 46.9 ± 8.6 | 0.032 |
| Trait anxiety | 47.4 ± 10.3 | 46.4 ± 10.3 | 46.9 ± 9.7 | 0.027 |

**Note:**
BF, body fat; BMI, body mass index.

and determined using the Akaike information criterion (AIC). A significance level of 5% was adopted for all analyses.

# RESULTS

## Physical characteristics macronutrient intake, and micronutrient intake

Table 1 summarizes the physical characteristics and State anxiety of the athletes. In Short sleeper group, both state and trait anxiety were found to be statistically higher than in the Long sleeper group ($p < 0.05$, for both, as described in Table 1).

Table 2 presents a summary of macronutrients intake among the athletes. Notably, in the Short sleeper group, daily energy intake, as well as protein and carbohydrate consumption, were statistically lower compared to the Long sleeper group ($p < 0.05$, for all, see Table 2). The percentage of protein for total energy intake was 17.2 ± 0.02% in the Long sleeper group and 17.0 ± 0.02% in the Short sleeper group with no significant difference detected (F = 0.2032, $p > 0.05$). Likewise, the percentage of fat for total energy intake was 30.8 ± 0.04% in the Short sleeper group and 30.2 ± 0.05% in the Long sleeper group with no significant difference detected (F = 0.290, $p > 0.05$). The percentage of carbohydrate (CHO) for total energy intake was 50.7 ± 0.06% in the Long sleeper group and 50.2 ± 0.04% in the Short sleeper group with no significant difference detected (F = 0.1514, $p > 0.05$).

Table 3 provides a summary of micronutrients intake among the athletes. In Short sleeper group, the intake levels of vitamin B1, vitamin B2, and vitamin C were statistically lower than those in the Long sleeper group ($p < 0.05$, for all, see Table 3).

## Verification of factors affecting menstrual irregularities

The GLM analysis revealed a significant association between an increase in state anxiety and the prevalence of menstrual irregularity (z = 2.000, $p = 0.048$, see Table 4). Additionally, a reduction in sleep duration did not exhibit a significant influence on the prevalence of menstrual irregularities (z = 1.493, $p = 0.136$, see Table 4). Subsequently, we observed that dietary energy intake did not have a significant impact the prevalence of menstrual irregularities (z = 1.029, $p = 0.303$, see Table 4).

**Table 2  Mean values of energy and macronutrient intakes (Mean ± SD).**

|  | All participants | Long sleeper | Short sleeper | *p* |
|---|---|---|---|---|
| Energy (kcal) | 2,386.0 ± 511.4 | 2,435.0 ± 512.8 | 2,160.3 ± 449.4 | 0.014 |
| Protein (g) | 102.2 ± 24.0 | 104.4 ± 23.2 | 92.3 ± 25.3 | 0.042 |
| Fat (g) | 80.3 ± 20.4 | 81.7 ± 20.6 | 73.9 ± 18.2 | 0.079 |
| CHO (g) | 302.6 ± 78.4 | 309.6 ± 80.6 | 270.4 ± 58.5 | 0.010 |

Note:
    CHO, carbohydrate.

**Table 3  Mean values of micronutrients intakes (Mean ± SD).**

|  | All participants | Long sleeper | Short sleeper | *p* |
|---|---|---|---|---|
| Ca (mg) | 734.8 ± 324.2 | 751.8 ± 341.0 | 657.4 ± 222.4 | 0.105 |
| Fe (mg) | 12.6 ± 4.0 | 12.7 ± 3.9 | 12.2 ± 4.5 | 0.637 |
| VA (μg RAE) | 930.5 ± 988.1 | 971.7 ± 1,055.9 | 742.4 ± 566.7 | 0.149 |
| $VB_1$ (mg) | 1.6 ± 0.7 | 1.6 ± 0.7 | 1.4 ± 0.4 | 0.011 |
| $VB_2$ (mg) | 1.8 ± 0.7 | 1.9 ± 0.7 | 1.6 ± 0.6 | 0.023 |
| VC (mg) | 160.0 ± 89.6 | 166.3 ± 95.1 | 130.9 ± 50.1 | 0.014 |
| Fiber (g) | 16.5 ± 5.8 | 16.7 ± 6.2 | 15.9 ± 3.5 | 0.420 |

Note:
    Ca, calcium; Fe, iron; VA, vitamin A; $VB_1$, vitamin $B_1$; $VB_2$, vitamin $B_2$; VC, vitamin C.

**Table 4  Variables and estimates of each variable of the multiple logistic models.**

| Variables | Estimate ± S.E. | z value | *p* value | O.R. | 95%CI for O.R. |
|---|---|---|---|---|---|
| State anxiety | 0.029 ± 0.015 | 2.000 | 0.048 | 1.030 | [1.000–1.059] |
| Short sleep duration | 0.450 ± 0.302 | 1.493 | 0.136 | 1.569 | [0.845–2.780] |
| Energy intake | 0.000 ± 0.000 | 1.029 | 0.303 | 1.002 | [0.998–1.006] |

Note:
    O.R., odds ratio.

## DISCUSSION

The present study highlights that among female athletes, sleep deprivation significant increased both state and trait anxiety (Table 1), as well as affecting energy intake (Table 2), particularly protein and carbohydrate intakes. However, GLM analysis indicated that while sleep deprivation did not directly influence the prevalence of menstrual irregularities, state anxiety emerged as a significant factor impacting the prevalence of menstrual irregularities. This suggests a potential pathway wherein sleep deprivation might elevate state anxiety levels, consequently indirectly contributing to an increase the probability of menstrual irregularities.

A previous population-based study demonstrated the positive associations between mental health problems, inadequate sleep duration, and the probability of menstrual irregularities (*Kim et al., 2018*). It suggested that a combination of short sleep duration and high levels of psychological stress, depressive mood, or suicidal ideation was associated with an increased the probability of menstrual irregularity and amenorrhea. Sleep is an

essential component of daily life and closely intricately linked with overall health (*Sabo et al., 1991*; *Tsuno, Besset & Ritchie, 2005*). Consequently, insufficient sleep can significantly impact on mental health (*Bushey, Tononi & Cirelli, 2011*).

In addition to its impact on mental health, sleep also plays a crucial role in physiological processes. Sleep supports cell growth and booths immune function are enhanced, highlighting its importance for overall well-being. Melatonin, the primary hormone secreted during sleep, regulates biorhythms (*Turek & Gillette, 2004*). However, evidence regarding the relationship between sleep deprivation and mental health, such as psychological anxiety, within the population of athletes. Further studies are needed in the near future to clarify these relationships specifically among athletes using a device that can estimate sleep duration.

Previous studies have observed a relationship between short sleep duration and increased total energy intake in general populations (*Tu et al., 2012*; *Ohida et al., 2001*; *Kim, DeRoo & Sandler, 2011*). The mechanisms underlying the association between sleep duration and food intake appear to be multifactorial. They include differences in the appetite-related hormones leptin and ghrelin, hedonic pathways, prolonged intake duration, and altered timing of intake. However, previous epidemiologic studies have shown that the association between short sleep duration and higher food intake (*Dashti et al., 2015*).

In contrast, the present study found statistically significant trends or differences in macronutrients, including energy, carbohydrates, and protein, associated with shorter sleep duration. A previous study reported that lack of sleep can increase ghrelin levels by up to 14.9% (*Taheri et al., 2004*), and reduce leptin secretion in large general population (*Vendrell et al., 2004*). Nevertheless, our findings showed that the Long sleeper group exhibited higher dietary energy intake compared to the Short sleeper group (Table 2). These outcomes observed in the athletes contrast with those reported in previous studies conducted among general population (*Taheri et al., 2004*, *Vendrell et al., 2004*). Athletes, typically more physical active and often consuming fewer calories relative to their energy expenditure, may exhibit different trends compared to the obese and general population, which tends to be less physically active and consume more relative to their energy expenditure. However, these findings will warrant further investigation in future studies.

There are several potential limitations that should be acknowledged in the present study. Firstly, data on menstrual cycles and sleep duration were obtained potentially introducing recall bias. The actual sleep duration is unknown as the survey relied on a questionnaire. Furthermore, this study did not account for the time of naps. The STAI, widely used clinically and featuring a well-defined clinical cutoff line (>40), was utilized to assess psychological anxiety (*Grant, McMahon & Austin, 2008*) revealing that athletes in our study had high levels of psychological anxiety. In fact, 78 of 128 athletes had state anxiety over 40, and 94 of 128 athletes had trait anxiety over 40. The results of the present study were consistent with the results of previous studies suggesting that the athletes had higher scores compared to the general population (*Gulliver et al., 2015*). The findings of the present study also indicate that female athletes may experience menstrual irregularities in the presence of heightened psychological anxiety. These are also consistent with the

previous studies (*Miyamoto, Hanatani & Shibuya, 2021b*, *2022*). In conclusion, this study provides novels insights suggesting that sleep deprivation might increase directly state anxiety and then indirectly increase the provability of menstrual irregularities. Therefore, it was suggested that decreased sleep duration may be related to mental health and menstrual irregularities among the three main characteristics of female athletes: mental health, menstrual irregularities, and decreased bone density.

The STAI, widely used clinically and featuring a well-defined clinical cutoff line (>40), was utilized to assess psychological anxiety (*Grant, McMahon & Austin, 2008*) revealing that athletes in our study had high levels of psychological anxiety. In fact, 78 of 128 athletes had state anxiety over 40, and 94 of 128 athletes had trait anxiety over 40. The results of the present study were consistent with the results of previous studies suggesting that the athletes had higher scores compared to the general population (*Gulliver et al., 2015*). These findings of the present study also indicate that female athletes may experience menstrual irregularities in the presence of heightened psychological anxiety. These are also consistent with the previous studies (*Miyamoto, Hanatani & Shibuya, 2021b*, *2022*).

In conclusion, the results of the presents study provide novels insights suggesting that sleep deprivation might directly increase state anxiety and indirectly affect the prevalence of menstrual irregularities. Hence, decreased sleep duration might be related to mental health issues and the prevalence of menstrual irregularities both significant concerns among female athletes. Future studies will play a crucial role in further elucidating how sleep patterns impact the health and well-being of female athletes.

### Funding
The authors received no funding for this work.

### Competing Interests
Kenichi Shibuya is an Academic Editor for PeerJ. Kenichi Shibuya and Mana Miyamoto are the Special Issue Editors.

### Author Contributions
- Mana Miyamoto conceived and designed the experiments, performed the experiments, analyzed the data, prepared figures and/or tables, authored or reviewed drafts of the article, and approved the final draft.
- Kenichi Shibuya conceived and designed the experiments, analyzed the data, prepared figures and/or tables, authored or reviewed drafts of the article, and approved the final draft.

### Human Ethics
The following information was supplied relating to ethical approvals (*i.e.*, approving body and any reference numbers):

The study protocol received ethical approval from the Ethics Committee of Niigata University of Health and Welfare (Approval #18147-19032).

## Data Availability

The raw measurements are available in the Supplemental File.

## Supplemental Information

Supplemental information for this article can be found online at http://dx.doi.org/10.7717/peerj.16976#supplemental-information.

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
