# Peer review of "Sleep duration has a limited impact on the prevalence of menstrual irregularities in athletes: a cross-sectional study"

_PeerJ, doi:10.7717/peerj.16976_

## Round 0.1 · original submission · Major Revisions

As you can see, the reviewers have offered constructive feedback that I believe will greatly assist you in revising your manuscript. I kindly request that you provide comprehensive responses to each comment from the reviewers.

Reviewer 1 ·

Basic reporting

It is suggested that authors seek more current references on the topic and use works with female athletes instead of references with men, the general population, and obese people.

Experimental design

INTRODUCTION
I suggest the authors delve deeper into the sports theme, describing the impact of the investigated variables (sleep, anxiety, and menstrual irregularities) on important aspects in the elite sports context, such as recovery and sports performance. In addition to also describes the impact of elite sports practice on sleep, anxiety levels, nutrition, and the menstrual cycle. Although the introduction is well structured, the authors do not concisely address the topic of sport, thus not contextualizing the reason for the investigation with athletes.
I suggest justifying the importance of carrying out this research in the sports field. What impact do the authors expect from carrying out this research on women's sports? Why is this study important?
I suggest that the authors finish the introduction by describing the research problem as well as its objective so that the reader has the possibility of understanding what the research is proposing. This information is not clear.

METHODOLOGY
The methods need to be described in more detail, especially about the instruments used. Regarding sleep and menstrual irregularities, it appears that the authors used questionnaires that have not been validated, in this sense, they must describe in detail the questions that were asked to the athletes so that the research can be replicated. Furthermore, it is suggested that they include the structured questionnaire as a supplementary file.

Validity of the findings

Poor conclusion, it is suggested that the authors rewrite the conclusion, also pointing out the practical applicability of their research for coaches and health team members who work with female athletes.

Additional comments

RESULTS
- The title of Table 1 refers to the physical and anxiety characteristics of the athletes, however, in the description of the results the authors only refer to the physical characteristics. It is suggested that authors review the description of the results related to Table 1 as well as the table title.
Furthermore, it is suggested that authors, when citing Table 1, give a brief description of the results presented in the table without repeating information, and only after this description, begin the description of the results in Table 2, as they are confusing for the reader.
- It is suggested that authors rename the group names, as the way it is described is too broad, the suggestion is to change it to “Short sleeper” and “long sleeper”

DISCUSSION
- I suggest removing age from the first paragraph of the discussion, as it was not a predominant factor in the objective or the statistical analyses. It is suggested to start the discussion by addressing the main result found in the research.
- It is suggested to cite studies carried out with obese women instead of studies with men.
- I suggest describing the main physiological functions sleep performs and the importance of sleep for elite athletes. Because despite sleep being one of the main variables investigated in the research, the topic of sleep and its importance and functions were little or not described.
- It is suggested that authors deepen the discussion on the main result found in the research. The way the discussion is described, the authors draw attention to a non-significant result, considering that the study reported high levels of anxiety appear to be related to menstrual irregularities.
- Furthermore, I suggest that the authors seek to discuss their results with studies with samples of female athletes. Also addressing characteristics related specifically to the sporting context and not the general population and obese people.

Annotated reviews are not available for download in order to protect the identity of reviewers who chose to remain anonymous.

·

Basic reporting

I appreciate the opportunity to review this article. Research involving female athletes lacks investigation; therefore, I consider the merit of the work relevant. Below, I sent some suggestions that I thought were pertinent.

Abstract
- two tires? Is this right?
- In the background, two groups are divided, Tier 3 and Tier 4. Is it necessary? This division does not appear anywhere else in the abstract.
- Where does it say, “Menstrual status and sleep duration were assessed using a questionnaire”, were they also assessed daily for 3 days? Also, was a questionnaire or diary used?
- If you have space, I suggest including the statistical analysis used.
- I suggest reviewing the abstract method, as it lacks information to understand how the study was developed.
- In the results when presented about age, the division into groups was not in relation to the competitive level? Or was it because of age? It wasn't clear, I suggest reviewing it.
- What is IF?
- In the description of the results, I suggest just putting p = 0.04 instead of P-value: 0.04.
- In the results, what is the p and r value of the correlation? Is it a positive or negative correlation?

Introduction
- Lines 23 and 24 - Isn't that what the study above discusses? It seems like it got repetitive
- Line 25 - I suggest using sleep restriction instead of sleep deprivation. If you do, review the entire article.
- Line 27 - Standardize whether it is mental or psychological anxiety.
- Line 34 - I don't think it would be, in other words, but rather, another hypothesis of the study.
- Lines 36 to 38 - This part lacks reference and seems more like a justification, which could come before the objectives and hypotheses.
- Lines 38 to 40 - It was repetitive; I suggest deleting it. The objective was presented twice in two consecutive paragraphs.

Methods
- Participants - Were there inclusion and exclusion criteria established?
- Line 49 - Was being on these teams a prerequisite, or was it out of convenience? Was there any intention of the researchers to select these two modalities?
- Line 68 - As self-reported, whether the scale made this assessment?
- Line 76 - I suggest describing when it was evaluated instead of citing other studies. Was the questionnaire reliability tested? Another suggestion is to describe the scoring criteria. What does the value 43.2 on state anxiety that appears in Table 1 represent?
- Line 79 – And how was it evaluated? For one or more days? Furthermore, I suggest making it clear that it was a souvenir.
- Line 84 – Fix formatting “.,”
- Lines 85 and 85 - Repetitive.. this information already exists at the beginning of the paragraph.
- Lines 86 to 88 - This should go up there when talking about the questionnaire. As it is, the reading doesn't flow.
- Line 89 - What questionnaire is this? Why was it only applied once? Wouldn't it be a sleep diary that the individual fills out daily?
- Lines 74 to 91 – This part is very long and could be divided into two or three paragraphs detailing each instrument used.
- Line 99 – ANOVA does not analyze the impact of sleep on anxiety and intake, right? For example, wouldn't it be an unpaired T-test to compare anxiety between the two groups (>6h and <6h)?

Results
Tables – Check the design, as there are some small squares at the top and bottom of the table.
- Line 112 – Regarding table 1, it summarizes other information in addition to physical characteristics, for example, anxiety.
- Line 114 - What is Cohen’s f? This is not in the statistical analysis.
- Line 124 - And the results in table 3? The vitamins were significantly different, I suggest addressing this topic.
- Line 127 - Would it be possible to create a figure?

Discussion
- Line 144 - statistically significant trends or differences? I suggest defining.
- Line 146 - I suggest including another study with a more similar population.
- Line 159 - Check reference formatting.
- Line 177 – About device.. I suggest giving examples, such as actigraphy.

Experimental design

no comment

Validity of the findings

no comment

Additional comments

no comment

Reviewer 3 ·

Basic reporting

I think this paper has reported some very interesting findings regarding the factors related to menstrual irregularity in young elite female athletes and their degree of influence, but I am curious about the following;

Experimental design

1) I think more detailed information about the participants' sports disciplines (e.g., number of people in each discipline) would be needed.

2) What anxieties did the participants respond to on the STAI? In the case of elite female athletes, there are a variety of anxieties (sport performance itself, health/physical condition, future, etc.).

3) I think it is necessary to describe at what point in the competitive season the measurements were taken and why the measurements were taken at that time.

4) Is it possible that some participants lived in a dormitory and were served the same meals? I think that information from meal providers is necessary.

Validity of the findings

1) I understand that anxiety affects menstrual irregularity, but on the other hand, is it possible that menstrual irregularity increases anxiety?

---

## Round 0.2 · accepted · Accept

Thank you for addressing the reviewers' concerns.

Reviewer 1 ·

Basic reporting

Review English, work continues with grammatical errors

Experimental design

the authors complied with the requested modifications

Validity of the findings

no comment

Additional comments

no comment

·

Basic reporting

No comment.

Experimental design

No comment.

Validity of the findings

No comment.

Additional comments

The authors responded to all the suggestions and I consider the article suitable for publication.

Reviewer 3 ·

Basic reporting

I confirmed that the manuscript has been appropriately revised in response to my comments.
I have nothing further to point out.

Experimental design

No comment

Validity of the findings

No comment

Additional comments

No comment